# 2M-AF: A Strong Multi-Modality Framework For Human Action Quality Assessment with Self-supervised Representation Learning

## ABSTRACT

Human Action Quality Assessment (AQA) is a prominent area of research in human action analysis. Current mainstream methods only consider the RGB modality which results in limited feature representation and insufficient performance due to the complexity of the AQA task. In this paper, we propose a simple and modular framework called the Two-Modality Assessment Framework (2M-AF), which comprises a skeleton stream, an RGB stream and a regression module. For the skeleton stream, we develop the Self-supervised Mask Encoder Graph Convolution Network (SME-GCN) to achieve representation learning, and further implement score assessment. Additionally, we propose a Preference Fusion Module (PFM) to fuse features, which can effectively avoid the disadvantages of different modalities. Our experimental results demonstrate the superiority of the proposed 2M-AF over current state-of-the-art methods on three publicly available datasets: AQA-7, UNLV-Diving, and MMFS-63.

## CCS CONCEPTS

• **Computing methodologies**;

## KEYWORDS

Action Quality Assessment, skeleton, RGB

## 1 INTRODUCTION

Action quality assessment (AQA) models have witnessed a surge in recent years due to their importance in various applications [1, 2, 7, 12]. Compared with traditional tasks such as classification or segmentation [5, 8], the AQA task presents more challenges. Specifically, human actions for the AQA task usually have the same background and action type but significantly different degrees of completion, which cause the networks for conventional action tasks with inadequate discrimination power in the AQA task [23].

To address this challenge, researchers have explored improvements in feature capture [19, 23] and regression modules [28, 29]. However, achieving a breakthrough on challenging datasets remains elusive, which showcases the limitations of RGB modality. RGB data contains rich environmental and colour information, but lying in the limited representation of spatial relationships and sensitivity to external factors, the RGB methods are hard to capture the important features of human actions. The result reflected in

Permission to make digital or hard copies of all or part of this work for personal or classroom use is granted without fee provided that copies are not made or distributed for profit or commercial advantage and that copies bear this notice and the full citation on the first page. Copyrights for components of this work owned by others than the author(s) must be honored. Abstracting with credit is permitted. To copy otherwise, or republish, to post on servers or to redistribute to lists, requires prior specific permission and/or a fee. Request permissions from permissions@acm.org.

*ACM MM, 2024, Melbourne, Australia*

© 2024 Copyright held by the owner/author(s). Publication rights licensed to ACM.
ACM ISBN 978-x-xxxx-xxxx-x/YY/MM
https://doi.org/10.1145/nnnnnnn.nnnnnnn

the AQA task is that the RGB modality struggles to achieve better performance in some environmentally irrelevant actions (e.g. gym, figure skating) because it is not only highly redundant in its features, but also more sensitive to subtle motion variations. To solve this problem, recent methods [28, 29] had to use complex regression structures, which led to complex calculations. Based on the above problem, we consider the introduction of another modality to ameliorate the defects of RGBs.

As a comparison to the RGB modality, the skeleton modality showcases different advantages for video tasks, including its intuitive spatial representation and robust feature specificity [6, 27]. Futhermore, the skeleton-based models offer significantly fewer parameters and faster computational speed than RGB ones [4]. But the flaw in skeleton data is that all environmental information is neglected, some of which may be crucial (e.g. splashes in diving sports). Therefore, designing a method to harness the advantages of both modalities will be effective. However, previous mainstream AQA researches seem to have little interest in skeleton as it seems not perform as well as RGB in unimodal states. Consequently, the purpose of our work is to **design a competitive skeleton-based network and further build a multi-modality, simple and efficient AQA framework**.

In this paper, we propose a Self-supervised Mask Encoder Graph Convolution Network (**SME-GCN**) to achieve representation learning for skeleton data. SME-GCN separately preserves the dynamic and temporal characteristics of skeleton data by developing two masking strategies to achieve an additional emphasis on the motion and obtain high-quality contrastive sequences. Building upon these masked data, SME-GCN then employs a siamese network and contrastive loss for self-supervised learning. By simply attaching a regression structure, SME-GCN improved the performance of the skeleton model on the AQA task, which reflects the excellent utilization of skeleton characteristics.

For the feature regression in action-based multi-modality methods, a common idea is to share the information in one or more layer(s) [13, 15]. These methods tend to introduce more extra computation in multi-modality calculations. In the AQA task, since almost each action has at least one modality capable of achieving good performance, we consider the feasibility of a new fusion approach that trains each sample to automatically select a more appropriate result from two modalities. Therefore, we design Preference Fusion Module (**PFM**) to choose the modality which is more suited for different action samples. Combining our SME-GCN, PFM and a simple RGB stream (I3D), we propose the Two-Modality Assessment Framework (**2M-AF**). It is worth noting that multiple backbones and regression losses can be configured within the modular 2M-AF, which exhibits the flexibility of our work. Our experimental results show that our 2M-AF has significantly surpassed the previous state-of-the-art methods on AQA-7, UNLV-Diving and MMFS-63 datasets.

The key contributions we make are as follows:

1. We propose a skeleton-based self-supervised representation learning network named SME-GCN which incorporates a unique masking strategy designed specifically for the characteristics of skeleton data. To the best of our knowledge, we are the first to apply self-supervised representation learning to the AQA task and achieve performance comparable to mainstream RGB approaches.

2. We propose a simple and efficient PFM module which can effectively avoid the disadvantages of different modalities and enhance the quality of multi-modality fusion without excessive hyperparameter tuning.

3. By combining our SME-GCN, PFM module and an RGB stream, we propose a Multi-Modality Action Quality Assessment Framework called 2M-AF. Our experimental results show that 2M-AF outperforms the State-of-the-art methods on three competitive datasets, AQA-7, UNLV-Diving and MMFS-63.

## 2 RELATED WORKS

### 2.1 Action quality assessment

RGB-based methods are the primary approaches used in current AQA tasks. The traditional methods [19, 21, 23, 26] utilize feature extractors like C3D or I3D to obtain features from videos and the final scores are generated through an LSTM or MLP regress module. However, due to the high requirements of feature discriminability in the AQA task, the previous methods have failed to achieve satisfactory results. Thus, feature regress modules emerge as a new research branch. C3D-AVG-MTL [20] adopts multi-task learning, simultaneously performing assessment, text generation and recognition tasks, but mutual interference by multi-task leads to an undesirable performance. CoRe-based methods [28, 29] utilize contrast learning to reduce the difficulty of scoring by comparing samples of the same class. However, training contrastive learning methods relies on additional category annotations and extra storage spaces. Recently, attention mechanisms have been utilized to highlight assessment-related information in features. TSA-Net [24] proposed a spatial-temporal attention module called Tube to reduce the detrimental effects of redundant information. Overall, researchers have done considerable work on RGB data, but they have still not made ideal progress, which has led us to consider adding extra modality information to the AQA model.

### 2.2 Skeleton-based Self-supervised representation learning

Self-supervised representation learning aims to obtain high-quality features from unlabeled samples. Early methods generated features through pretext tasks [16, 25]. Then, self-supervised learning rapidly progressed and achieve remarkable performance, which is comparable to supervised learning. Some methods [3, 11] utilize contrastive learning by constructing pairs of samples to learn the correlations between samples and enable the backbone network to acquire rich discriminative representations. Building on this foundation, in the past two years, some methods [9, 32] have started to explore the application of self-supervised learning in skeleton-based action recognition but fail to consider the unique spatial-temporal relationship of joints. After that, MAE [10], which adopts pixel masking to reconstruct images using encoder-decoder

structures, lets researchers realize the importance of masking. As an improvement, PSTL [30] proposes a spatial-temporal masking strategy to generate pairs of skeleton samples. Overall, the masking strategy is primarily aimed at incorporating variability information while preserving key features. However, due to the complexity and atypical nature of AQA actions, masking at the skeleton dimension can easily compromise the fine details of the actions themselves. Therefore, the transfer learning of methods like PSTL is limited in terms of performance for the AQA task, which promotes us to build a self-supervised network more fitting our assignment.

## 3 METHODS

### 3.1 Self-supervised Mask Encoder Graph Convolution Network

As Figure 1 (4) shows, the original skeleton sequence will be transformed into two new sequences using the frame-level masking strategies. Then, the two sequences will be processed by the encoder with shared weights. Finally, the model will achieve self-supervised learning using contrast loss.

**Masking strategy.** A 3D human skeleton sequence can be denoted as $x \in \mathbb{R}^{C \times T \times V}$, which has C channels, T frames, and V joints. As the input of SME-GCN, we define skeleton sequences as $x^s \in \mathbb{R}^{N \times C \times T \times V}$, where N is the batch size, and $x_i^s \in \mathbb{R}^{C \times T \times V}$ represents the $i^{th}$ sequence in $x^s$.

In self-supervised tasks, the first and most crucial step is to construct partial skeleton data, which should retain sufficient key information to preserve the original characteristics of the actions. Our masking strategy pay extra attention to the spatial-temporal properties of the skeleton. Our variance mask module aims to focus on the shift within actions. First, we subtract the coordinates of each joint from the coordinates of the central joint, which provides a standardized measure:

$$x_{i,:,:,:}^s = x_{i,:,:,:}^s - x_{i,:,:,m}^s, \tag{1}$$

where $x_{i,:,:,m}^s$ denotes the centre joint $m$ of $x_i^s$, we recommend selecting the joint closer to the human centre as the central joint (for HRNet, we choose $m = 8$). Next, we define the alteration degree $D_i(t)$ to obtain a measure of change for the $t^{th}$ frame of the $i^{th}$ skeleton:

$$D_i(t) = \sum_{c=1}^{C} \sum_{v=1}^{V} \sum_{\substack{j=t-k \\ j \in [0,T-1]}}^{t+k} (x_{i,c,j,v}^s - \overline{x}_{i,c,t,v}^s)^2, \tag{2}$$

where $k$ is a hyperparameter that represents the range of perception, and $x_{i,c,j,v}^s$ indicates the $c^{th}$ channel, $t^{th}$ frame and $v^{th}$ joint of $x_i^s$, $\overline{x}_{i,c,t,v}^s$ denotes the average of the data from the $(t-k)^{th}$ to $(t+k)^{th}$ frames (if the index is a valid number). By calculating the variances between frames and their surrounding $k$ frames, we can mask out skeleton frames which have lower change. We sort the index list of the $x_{i,t}^s$ in ascending order based on $D_i(t)$. As we define the mask rate $r_m$, we acquire the $(r_m)T^{th}$ index and label its $D_i(t)$ as $\alpha$. The variance masked sequence $\hat{x}_i^s$ is represented as:

**Figure 1: The overall architecture of 2M-AF. (1) The skeleton stream. HRNet [22] is utilized to estimate skeleton data. (2) The RGB stream. (3) The diagram of our PFM regress module. $C$ represents concatenate, each dot indicates a prediction of a sample, and Reg represents the regress loss (MSE+MAE is set in our work). The central classifier optimizes the stream towards predicted results that are closer to the ground-truth scores. In inference, the classifier will directly choose the final prediction from the outputs of two streams. (4) The structure of our SME-GCN for representation learning. The diagonal lines represent masked frames. GAP in the GCN encoder denotes global average pooling.**

$$\hat{x}_{i,t}^{s} = \begin{cases} 0 & if \quad D_i(t) < \alpha \\ x_{i,t}^{s} & else \end{cases}. \qquad (3)$$

Our strategy obtains strong data while preserving the dynamic characteristics of the original data. In contrast, the temporal mask strategy is primarily used to preserve the temporal characteristics of the data. Therefore, the new sequence $\widetilde{x}_i^{s}$ is formed by applying the interval mask based on $r_m$. In summary, we have generated two samples which are preserving key characteristics from the original data. This lays a crucial foundation for improving the performance of our self-supervised learning.

**Encoder and loss function.** As a modular approach, various skeleton encoders can be configured with SME-GCN. After conducting extensive experiments, we ultimately choose the graph convolutional part of CTR-GCN [4] and incorporate a fully connected layer as the encoder in SME-GCN. The sequences $\hat{x}_i^{s}$ and $\widetilde{x}_i^{s}$ will be passed into the weight-shared encoder to generate feature vector pair, $\hat{f}_i^{s} \in \mathbb{R}^{C^s}$ and $\widetilde{f}_i^{s} \in \mathbb{R}^{C^s}$ ($C^s$ is the number of the output channel). For ease of description, we merge $N$ pairs of $\hat{f}_i^{s}$

and $\widetilde{f}_i^{s}$ into $f^m \in \mathbb{R}^{2N \times C^s}$ by placing all $\widetilde{f}_i^{s}$ after $\hat{f}_i^{s}$. With similarity measured, InfoNCE [17] is adapted as the contrast loss of our method:

$$\mathcal{L}_{SME} = -\frac{1}{N} \sum_{i=1}^{N} log \frac{e^{sim(f_i^m, f_{N+i}^m)/\tau}}{\sum_{k=1, k \neq i}^{2N} e^{sim(f_i^m, f_k^m)/\tau}}, \qquad (4)$$

where $sim(.,.)$ indicates the product of the $l_2$ normalised inputs and $\tau$ denotes a constant temperature parameter. As a robust method, our SME-GCN can be utilized as the skeleton backbone for multiple human action tasks. In addition to our main focused AQA task in this paper, we also explored the effectiveness of SME-GCN on recognition tasks in our experiments.

## 3.2 Multi-Modality Assessment Framework

**SME-GCN for AQA.** By appending a linear regress module on top of the encoder and then finetuning the entire network on the target AQA dataset, we build a skeleton-based network for the AQA task. The overall architecture of our 2M-AF is shown in Figure 1. In the skeleton stream, the input frames are sent into HRNet [22] to estimate skeleton data. The pretrained SME-GCN is utilized as

the backbone of the skeleton stream to obtain the skeleton feature $F^s \in \mathbb{R}^{N \times C^s}$.

**The RGB stream of 2M-AF.** In the RGB stream, the input frames are divided into $n$ small clips ($n$ is adjusted based on the length of the dataset). Then the clips are sent into I3D ConvNets for extracting features. Finally, we calculate the average of the features to realize the RGB feature $F^r \in \mathbb{R}^{N \times C^r}$, where $C^r$ represents the number of the output channel.

**Preference Fusion Module.** Our Preference Fusion Module (PFM) consists of two regression heads and a selection head to process the RGB and the skeleton features, $F^r$ and $F^s$. (1) In the regression heads, $F^r$ and $F^s$ separately regress the predicted score, $S^r \in \mathbb{R}^N$ and $S^s \in \mathbb{R}^N$ by a fully connected layer. (2) The selection head is the core of PFM. We concatenate $F^r$ and $F^s$ and then send the merged feature to a fully connected layer and a Softmax layer to acquire a two-category vector $F^c \in \mathbb{R}^{N \times 2}$ to compare with the preference label $y^c$, which is formed by calculating the L1 distance between the predicted score of each stream and the ground truth score. $y^c$ can be presented as:

$$y_i^c = argmin\{(|S_i^s - y_i|), (|S_i^r - y_i|)\}, \qquad (5)$$

where $y_i^c$ is the $i^{th}$ sample of $y^c$, and $S_i^s, S_i^r$ denotes the predicted score of the $i^{th}$ skeleton sample and RGB sample, respectively. To enable backpropagation, our loss function $\mathcal{L}$ combines both regression and classification losses. Motivated by MTL-AQA [20], our regression loss $\mathcal{L}_s$ for the skeleton adapted L1 distance in addition to MSE:

$$\mathcal{L}_s = -\frac{1}{N} \sum_{i=1}^{N} \lambda_1 (S_i^s - y_i)^2 + \lambda_2 |S_i^s - y_i|, \qquad (6)$$

where $y_i$ is the ground truth score, and $\lambda_1$ and $\lambda_2$ is represented as the weights for the two metrics. The loss function $\mathcal{L}_r$ for the RGB stream is the same as $\mathcal{L}_s$.

$$\mathcal{L}_r = -\frac{1}{N} \sum_{i=1}^{N} \lambda_1 (S_i^r - y_i)^2 + \lambda_2 |S_i^r - y_i|, \qquad (7)$$

We calculate the loss function separately for both streams and then sum them up to obtain the loss of the regression part. For the classification, based on the preference label $y^c$ and the output vector $F^c$, we utilize cross-entropy loss $\mathcal{L}_c$ to iterate parameters:

$$\mathcal{L}_c = -\frac{1}{N} \sum_{i=1}^{N} y_i^c \cdot log(F_{i,0}^c) + (1 - y_i^c) \cdot log(F_{i,1}^c), \qquad (8)$$

where $F_{i,k}^c$ indicates the $i^{th}$ sample and the $k^{th}$ feature of $F^c$. During the training process, the complete loss function $\mathcal{L}$ is obtained by adding up the aforementioned losses:

$$\mathcal{L} = \mathcal{L}_s + \mathcal{L}_r + \mathcal{L}_c. \qquad (9)$$

In testing, PFM selects the scores based on the classification results as the final predicted score. As a result, PFM facilitates independent regression of the two streams and possesses the ability to select the more accurate result for each sample from outputs. Considering all the aforesaid work, our 2M-AF solution addresses the challenge of fusing multimodal networks with attribute differences.

# 4 EXPERIMENTS

## 4.1 Datasets

In this section, we conducted experiments on three datasets with different characteristics and frame lengths to demonstrate the generalizability of our framework. AQA-7 [19] is currently the most widely used AQA dataset, which also has the most diverse range of action categories. In contrast, UNLV-Diving [21] and MMFS-63 [14] have longer frame sequences and consist of actions from a single sport. Additionally, MMFS-63 offers a large number of variable-length frame sequences and pre-processed, high-quality skeleton data. Consequently, experiments on the above three datasets which have different scales, frame lengths and adaptability to two modalities are conducted to showcase the strong generalization ability of our framework. To maintain consistency with existing literature, Spearman's rank correlation coefficient is employed as the measurement standard. (The detailed experimental configuration and download link of the three datasets is written in the Appendix.)

**AQA-7.** The AQA-7 dataset comprises samples from seven actions. It contains 1189 videos with 103 frame lengths, of which 803 videos are used for training and 303 videos are used for testing. To ensure a fair comparison with other methods, we remove the trampoline category the same as the previous works and calculate the performance of the other six categories. To compare with the mainstream methods, Fisher's z-value [19] is used to measure the average performance across actions. In order to avoid potential bias towards certain categories in the results and demonstrate the performance of our method on a large-scale, multi-category dataset, we additionally trained our model using all six categories of AQA-7 data in comparing with the current leading method and the ablation study.

**UNLV-Diving.** The UNLV-Diving dataset contains 370 video clips with various diving actions, of which 300 videos are used for training and 70 videos are used for testing. The length of each video is 151 frames. According to the dataset description, we use the execution score of actions as the evaluation label.

**MMFS-63.** MMFS-63 is a large AQA dataset that consists of 63 classes of actions, which were collected from the World Figure Skating Championships. There are a total of 4915 samples for both skeleton and RGB data, of which 3959 samples are for training and 956 for testing.

**Evaluation Protocols.** To maintain consistency with existing literature, we employed Spearman's rank correlation coefficient, which ranges from -1 to 1 (with higher values indicating better performance), to evaluate the accuracy of our methods in predicting the score series compared to the ground-truth data. As the metrics, Spearman's correlation coefficient is defined as follows:

$$\rho = \frac{\sum_{i=1}^{N} (p_i - \overline{p})(q_i - \overline{q})}{\sqrt{\sum_{i=1}^{N} (p_i - \overline{p})^2 \sum_{i=1}^{N} (q_i - \overline{q})^2}} . \qquad (10)$$

Here $p_i$ and $q_i$ represent the ranking of the $i^{th}$ predicted score and sample score series, respectively. And $\overline{p}$ and $\overline{q}$ denote the average of the two score series respectively. $N$ is the length of the series.

| Methods | Diving | Gym Vault | BigSki | BigSnow | S. 3m | S. 10m | Ave | All |
|---|---|---|---|---|---|---|---|---|
| C3D-LSTM | 0.6047 | 0.5636 | 0.4593 | 0.5029 | 0.7912 | 0.6927 | 0.6165 | – |
| C3D-SVR | 0.7902 | 0.6824 | 0.5209 | 0.4006 | 0.4006 | 0.9120 | 0.6937 | – |
| JRG | 0.7630 | 0.7358 | 0.6006 | 0.5405 | 0.9013 | 0.9254 | 0.7849 | – |
| MUSDL | 0.8099 | 0.7570 | 0.6538 | 0.7109 | 0.9166 | 0.8878 | 0.8102 | – |
| CoRe | 0.8824 | 0.7746 | 0.7115 | 0.6624 | 0.9442 | 0.9078 | 0.8401 | – |
| TSA-Net | 0.8379 | 0.8004 | 0.6657 | 0.6856 | 0.9459 | 0.9334 | 0.8476 | – |
| DAE-MLP | 0.8420 | 0.7754 | 0.6836 | 0.7230 | 0.9237 | 0.8902 | 0.8258 | 0.8693 |
| DAE-CoRe | **0.8923** | 0.7786 | 0.7102 | 0.6842 | 0.9506 | 0.9129 | 0.8520 | 0.8757 |
| 2M-AF (ours) | 0.8715 | **0.8050** | **0.7276** | **0.7254** | **0.9555** | **0.9380** | **0.8662** | **0.8901** |

Table 1: Accuracy comparison against existing methods on the AQA-7 dataset, where 'Ave' represents the performance calculated by Fisher's z-value [19], and 'All' represents the comparison with the all six categories of AQA-7 data.

| Methods | Sp. Corr. |
|---|---|
| C3D-SVR [21] | 0.7800 |
| JRG [18] | 0.7630 |
| C3D-AVG-MTL [20] | 0.8383 |
| FALCONS [15] | 0.8453 |
| CoRe [28] | 0.8663 |
| DAE-CoRe [29] | 0.8477 |
| 2M-AF (ours) | **0.8838** |

Table 2: Accuracy comparison against existing methods on the UNLV-Diving dataset.

## 4.2 Comparison with the State-of-the-Art

We compare our method with the state-of-the-art AQA methods on all three datasets introduced above. As results are shown in Table 1, we achieved advantages in five out of the six single categories on AQA-7, and in the other specific single category, 'Diving', we are still close to the best result. As results are shown in Tables 1, 2 and 3, our 2M-AF achieves state-of-the-art performance with a large margin on the three datasets. The results demonstrate the excellent robustness of 2M-AF to different types, scales and frame lengths of data.

| Methods | Sp. Corr. |
|---|---|
| C3D-LSTM [21] | 0.5234 |
| C3D-AVG-STL [20] | 0.3831 |
| CoRe [28] | 0.7313 |
| DAE-MLP [29] | 0.5915 |
| DAE-CoRe [29] | 0.7548 |
| 2M-AF (ours) | **0.8744** |

Table 3: Accuracy comparison against existing methods on the MMFS-63 dataset.

## 4.3 Ablation study

To validate the performance of each module through ablation experiments, firstly, we discuss the significance of introducing multi-modality. Then, we analyze the competence of SME-GCN from the aspects of masking rate, the effectiveness of the masking strategy, and performance compared with other self-supervised methods. Lastly, we demonstrate the capability of PFM by comparing it with other fusion methods. The experiments on the AQA-7 dataset utilize all six categories (like **'All'** in Table 1).

**Comparing between different modalities.** To demonstrate the performance on different modalities, we discuss the comparison between the RGB modality, skeleton modality, simple modality fusion (using a fully connected layer after concatenating the features), and PFM on AQA-7, UNLV-Diving and MMFS-63 datasets. To ensure a fair comparison, we set the RGB modality to use the I3D ConvNet while the skeleton modality is uniformly implemented using CTR-GCN. As experimental results are shown in Table 4, (1) different datasets exhibit distinct inclinations towards modalities. Both the skeleton modality and RGB modality perform well on AQA-7 because of its diverse range of action categories, while on MMFS-63, the skeleton modality exhibits a clear advantage, and on UNLV-Diving, the RGB modality performs better. This reflects that different data exhibit significant differences in performance across different modalities, highlighting the importance of multi-modality methods in the context of the AQA task. (2) for the modality fusion, we observe that the direct fusion strategy performs not only lower than PFM but also than the single modality, especially when there is a large difference between the performance of the two modalities (on MMFS-63 and UNLV-Diving). This aligns with our previous analysis, as the vast information disparity, simply concatenating features and using fully connected layers is inappropriate.

**Configuration of the skeleton backbone.** To choose a better skeleton backbone, we compared several mainstream skeleton networks to select the most suitable skeleton backbone for our 2M-AF. For equal comparison, we added a single MLP layer on each skeleton network to regress and predict scores for all skeleton networks. The experimental results in 5 demonstrate that CTR-GCN is a more suitable skeleton backbone.

| Dataset | RGB | Skeleton | SF | PFM |
|---|---|---|---|---|
| AQA-7 | 0.8517 | 0.8696 | 0.8631 | **0.8744** |
| UNLV-Diving | 0.8557 | 0.7744 | 0.8279 | **0.8618** |
| MMFS-63 | 0.7713 | 0.8595 | 0.8071 | **0.8692** |

**Table 4: Comparisons of accuracies when utilizing different modalities on AQA-7, MMFS-63 and UNLV-Diving datasets. SF denotes simply fusion modalities by an FC layer.**

| Methods | Accuracy |
|---|---|
| ST-GCN | 0.8618 |
| 2S-AGCN | 0.8691 |
| EfficientGCN B4 | 0.8593 |
| PoseC3D | 0.8689 |
| CTR-GCN | **0.8696** |

**Table 5: Comparisons of accuracies when utilizing different skeleton backbones on the AQA-7 dataset.**

| Settings | $r_m$ | Sp. Corr. |
|---|---|---|
| baseline | – | 0.8696 |
| A | 0.38 | **0.8848** |
| B | 0.69 | 0.8757 |
| C | 0.84 | 0.8713 |

**Table 6: Ablation study when utilizing SME-GCN on AQA-7 dataset with different $r_m$.**

| Methods | Sp. Corr. |
|---|---|
| baseline (CTR-GCN) | 0.8696 |
| temporal+random | 0.8737 |
| variance+random | 0.8803 |
| AimCLR [9] | 0.8706 |
| PSTL [30] | 0.8699 |
| temporal+variance | **0.8848** |

**Table 7: Comparisons of accuracies when utilizing different contrastive strategies on the AQA-7 dataset.**

**The masking rate configuration of SME-GCN.** As a self-supervised work designed for the AQA task, we compare the performance of SME-GCN under different configurations of $r_m$ and only CTR-GCN backbone [4] (baseline in Table 6). Due to each video in AQA-7 consisting of 103 frames, we selected three sets of masking configurations $r_m = 0.84$, $r_m = 0.69$, $r_m = 0.38$ to retain 16, 32, and 64 frames of valid data. As experimental results are shown in Table 6, regardless of the value of $r_m$, SME-GCN outperforms the regular CTR-GCN, demonstrating the effectiveness of our self-supervised strategy. Additionally, the results are even better when $r_m = 0.38$. We believe that this level of masking introduces

sufficient variations to the data without compromising important features.

**Effects of our masking strategies for SME-GCN.** In Table 7, we conducted ablation experiments on different masking strategies of SME-GCN. (1) We first compared our own masking strategies. The 'random' strategy indicates masking random frames, 'temporal' denotes temporal masking and 'variance' represents variance masking. The masking rate is set as $r_m = 0.38$ considering the above experiment. The experimental results demonstrate that including each masking strategy improves the model's accuracy. (2) Next, we compared our SME-GCN with other skeleton-based self-supervised representation masking strategies. As the result is shown, with the same CTR-GCN backbone, all self-supervised strategies improve the performance of downstream AQA tasks. This result validates our analysis that self-supervised learning can address the limitations of skeleton models, while our method achieves superior results compared to other methods due to the emphasis on preserving crucial spatial-temporal information in the features.

| Settings | Methods | Sp. Corr. |
|---|---|---|
| A | MLP | 0.8631 |
| B | BPAN | 0.8488 |
| C | FALCONS | 0.8680 |
| D | PFM+MLP | 0.8728 |
| E | PFM | **0.8744** |

**Table 8: Comparisons of accuracies when utilizing different regression heads with multi-modality model on the AQA-7 dataset.**

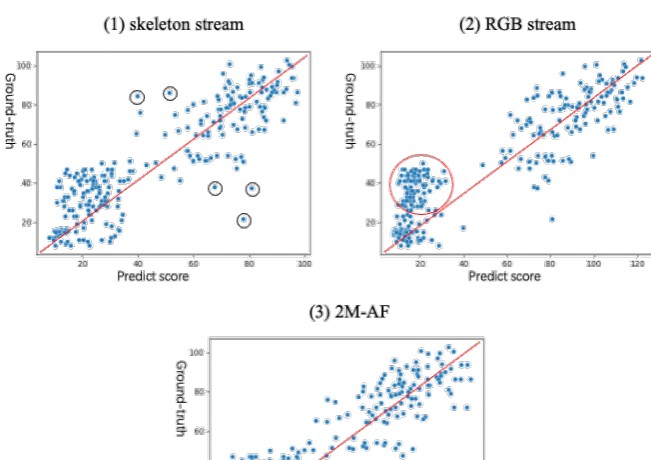

**Figure 2: A comparison of different methods in the scatter plot. Each point represents a sample in the test AQA-7 dataset. The red line denotes the perfect predictions.**

**The effects of PFM.** To validate the performance of PFM, we compared its performance with mainstream regression heads which combine RGB and skeleton modalities. All experiments were conducted on the AQA-7 dataset, using CTR-GCN backbone for the skeleton modality and I3D backbone for the RGB modality. The experimental results are shown in Table 8. Compared to settings A, B, C and E, PFM outperforms mainstream regression heads ([15, 31]), which indicates our method is superior to feature fusion approaches. Additionally, we conducted additional tests by adding a regression head that concatenates features and uses an FC layer to let PFM achieve selection from three outputs (Setting D). The experimental results show that this configuration does not improve performance. We are convinced this is due to the decrease in accuracy caused by the excessive redundancy of features.

**The configuration of loss functions for 2M-AF.** We compared several commonly used regression loss functions to determine the most suitable method for AQA tasks and our framework. Considering the three configurations in Table 9, we decided to use the combination of MAE (Mean Absolute Error) and MSE (Mean Squared Error) losses (Setting C) as the regression loss in 2M-AF.

| Settings | Losses | Sp. Corr. |
|----------|--------|-----------|
| A | MSE | 0.8699 |
| B | MAE | 0.8695 |
| C | MSE+MAE | **0.8744** |

**Table 9: Comparisons of accuracies when utilizing different loss functions for 2M-AF on the AQA-7 dataset.**

## 4.4 The efficiency of our method

As shown in Table 10, we compared the efficiency of our method with the State-of-the-art CoRe-based structure [28, 29] on the AQA-7 dataset. We first conducted a separate analysis of the efficiency of our skeleton network SME-GCN and found that it accounts for only about 1/5 Params and 1/10 FLOPs of the overall framework 2M-AF. This indicates that combining the skeleton modality and RGB modality will not result in an excessively large number of parameters. Additionally, we observe that our 2M-AF utilizes similar parameters but is more efficient in speed compared to the current mainstream solution, which means our 2M-AF no longer need complex regression structures but can achieve more outstanding performance.

| Methods | Params | FLOPs |
|---------|--------|-------|
| CoRe | 14.80M | 557.53G |
| SME-GCN | 3.72M | 15.3G |
| 2M-AF | 14.78M | 168.21G |

**Table 10: Efficiency comparison against CoRe.**

## 4.5 Visualization study

To have an intuitive understanding of the differences between our single skeleton stream, single RGB stream, and 2M-AF, we visualize the prediction results in the form of a scatter plot in Figure 2. From the graph, we can observe the following in the prediction of AQA-7: (1) The skeleton stream has more outlier points (such as the points in the black circle) compared with other plots. This means the skeleton model performs better on the majority of samples, but it produces noticeable errors on a small portion of samples, which may cause by the poor quality of the skeleton data for a few samples. (2) In contrast, the RGB stream shows many samples in the red circle deviations from the perfect predictions, suggesting its limited regression performance for dense samples. (3) Our 2M-AF successfully addresses the above two issues, which reflects the effectiveness of our PFM in selecting more accurate predictions from both streams.

## 4.6 Case study

To better illustrate the significant differences in the tendency of different samples towards modalities, we selected a few examples for demonstration in Figure 3. It can be observed that: (1) for diving actions 'a' and 'b', RGB data shows a clear advantage due to the significant influence of environmental features such as splashes on the quality assessment. As well, frames falling into the water are difficult for the skeleton to estimate, which makes the skeleton stream limited. (2) for skating actions 'c' and 'd', the skeleton stream demonstrates a noticeable advantage. This is attributed to the fact that skating actions rely more on spatial relationships which is suited for the skeleton methods, but the RGB methods will be badly subject to environmental factors.

## 4.7 Migration study

**The feature enhancement capability of SME-GCN.** In order to showcase the feature enhancement capability of SME-GCN, we additionally utilized the SME-GCN encoder with an added classification head for the action recognition task. Following our backbone CTR-GCN, we utilize the fusion results from four different skeleton modalities. The results are presented in Table 11. The experimental results demonstrate that SME-GCN achieved improvements in three branches of skeleton modalities and the final result, indicates that our self-supervised structure also plays a crucial role in feature enhancement for the classification task.

| Model | J | B | JM | BM | fusion |
|-------|-----|-----|-----|-----|--------|
| CTR-GCN | **89.93** | 89.09 | 86.62 | 84.53 | 92.4 |
| SME-GCN | 89.45 | **90.36** | **87.71** | **86.40** | **92.6** |

**Table 11: Comparisons of classification accuracies with CTR-GCN on the NTU RGB+D 60 X-sub dataset. J, B, JM, and BM represent joint, bone, joint motion, and bone motion skeleton modalities.**

We used the T-SNE method to visualize the features of the aforementioned classification task. Points with different colours represent different categories, and the results are shown in Figure 4. It

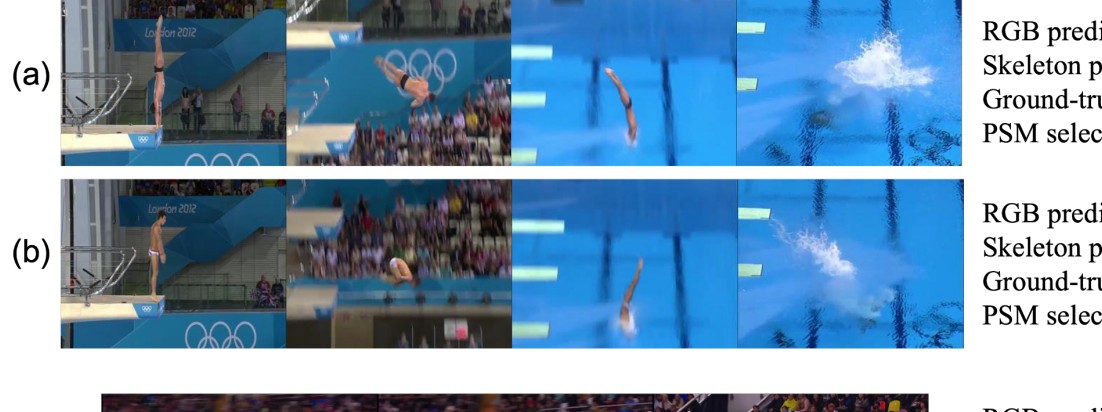

RGB prediction: 56.96
Skeleton prediction: 80.16
Ground-truth score: 52.00
PSM selection: RGB 56.96

RGB prediction: 78.46
Skeleton prediction: 57.95
Ground-truth score: 78.40
PSM selection: RGB 78.46

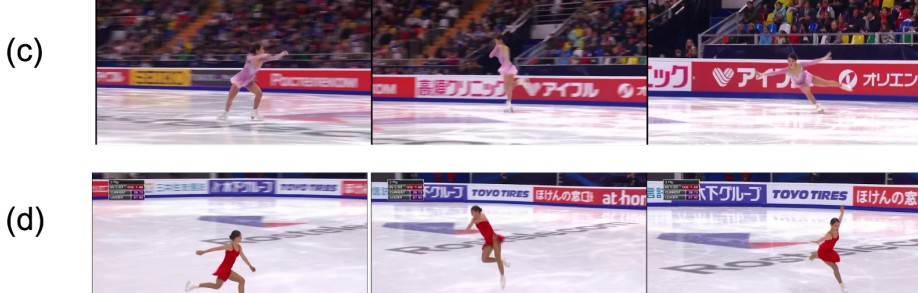

RGB prediction: 3.01
Skeleton prediction: 4.22
Ground-truth score: 4.38
PSM selection: Skeleton 4.22

RGB prediction: 4.52
Skeleton prediction: 3.09
Ground-truth score: 3.30
PSM selection: skeleton 3.09

**Figure 3: The case study on (a) (b) Diving and (c) (d) MMFS-63. Where (a) and (b) have the same difficulty (3.2). (c) and (d) utilizes the same '2Axel' category. The statistical results shown in (a) and (b) are the execution score * difficulty.**

can be observed that the SME-GCN on the right exhibits relatively better classification performance.

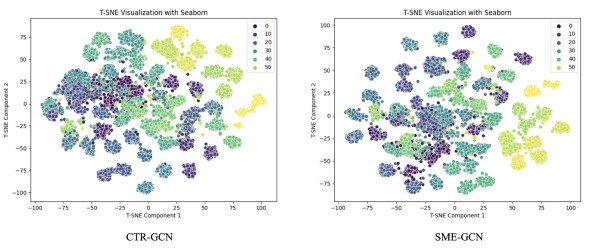

**Figure 4: Scatter plot by the T-SNE.**

**The feature enhancement capability by different mask rate on SME-GCN.** In order to showcase the feature enhancement capability of SME-GCN for multiple tasks, we additionally utilized the SME-GCN encoder with an added classification head for the action recognition task. The results are presented in Table 12. At different mask rates, SME-GCN achieves excellent performance at mask rates up to 0.25-0.5, and the performance degradation when the mask rate is higher is caused by the loss of too much information. The experimental results demonstrate that SME-GCN can achieve improvements on multiple tasks, which shows strong generalization capabilities.

**Table 12: Comparisons of classification accuracies in different mask rate on SME-GCN.**

| configurations | mask rate | frames | Acc |
|---|---|---|---|
| A | 0 | 64 | 92.4 |
| B | 0.25 | 48 | 92.5 |
| C | 0.5 | 32 | **92.6** |
| D | 0.75 | 16 | 91.7 |

## 5 CONCLUSION

This paper proposes a framework for the AQA task named 2M-AF, which is powerful for providing richer features and facilitating the utilization of multi-modality information. The framework comprises a self-supervised graph convolutional skeleton network called SME-GCN, an RGB backbone, and the proposed PFM module. The experimental results demonstrate that our method outperforms the existing approaches on the three challenging datasets. As a simple and flexible framework, we believe that the value of 2M-AF lies not only in its significant improvement in accuracy but also in its compatibility with various components, which provides endless possibilities for research built on the foundation of 2M-AF.

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
