# OpenReview forum: "2M-AF: A Strong Multi-Modality Framework For Human Action Quality Assessment with Self-supervised Representation Learning"
_acmmm.org/ACMMM/2024/Conference — MM2024 Poster_

### Official Review · Reviewer_LfTe · 2024-05-03

**Rating:** 3
**Confidence:** 2

**Summary:**

This paper designs a competitive skeleton-based network and further build a multi-modality, simple and efficient AQA framework. It also proposes a Self-supervised Mask Encoder Graph Convolution Network (SME-GCN) to achieve representation learning for skeleton data.

**Strengths:**

Overall, the paper is well-organized, and the technical contributions appear to be substantial.

**Limitations:**

1.It is recommended that the authors consider making the source code publicly available.
2.In the contributions, the authors state, "To the best of our knowledge, we are the first to apply self-supervised representation learning to the AQA task." However, in the related work, it is mentioned that "some methods have started to explore the application of self-supervised learning in skeleton-based action recognition." Therefore, I believe the authors' statement is not precise.
3.The paper compares different models for both performance and efficiency. Why were two different models chosen for comparison?
4.Ablation experiments aim to investigate the impact of adding or removing specific modules from the model on overall performance. However, in Section 4.3 of this paper, the ablation experiments discuss "The masking rate configuration of SME-GCN" and "The configuration of loss functions for 2M-AF."
5.There are some grammatical mistakes. For example, "Then, self-supervised learning rapidly progressed and achieve remarkable performance."

**Suitability:**

3

---

### Official Review · Reviewer_GUAf · 2024-05-13

**Rating:** 4
**Confidence:** 2

**Summary:**

The paper introduces a novel dual-modality framework, 2M-AF, which enhances the performance of human action quality assessment by integrating a skeleton stream with the traditional RGB-based approach. This work uses a Preference Fusion Module (PFM) to adaptively select results from both modalities, achieving state-of-the-art results on mainstream datasets.

**Strengths:**

1. The introduction of a skeleton stream alongside the traditional RGB stream is a significant enhancement, particularly given the self-supervised learning strategy employed to enable effective feature extraction from the skeleton data. This dual-modality approach effectively addresses the shortcomings of RGB-only systems in some environmentally irrelevant scenarios.
2. The method's superior performance on benchmark datasets confirms its efficacy and the practical value of integrating multiple modalities for action quality assessment.

**Limitations:**

### Major weaknesses
1. **Incomplete Review of Related Work:**
The paper discusses RGB-based action quality assessment methods but lacks a comprehensive review of existing skeleton-based approaches, which are also relevant to this research.
For instance, the study "Skeleton Based Action Quality Assessment of Figure Skating Videos[1]" should be discussed to highlight how the proposed method differs from or improves upon these approaches.
Including a broader range of related works would provide a more thorough context for your contributions and help clarify the novelty of your approach.

2. **About PFM：**
Table 8 lacks references to the methods BPAN and FALCONS, nor are they introduced in the PFM effectiveness subsection. Additionally, PFM essentially represents a hard selection for two-modality streams. How would the effect change if we opt for a soft selection by using the predicted scores from PFM for weighted summation of the two streams?

### Minor weakness
There is a typo in line 576: "results in 5" -> "results in Table 5"


[1] Li, Hui-Ying, et al. "Skeleton based action quality assessment of figure skating videos." 2021 11th International Conference on Information Technology in Medicine and Education (ITME). IEEE, 2021.

**Suitability:**

3

---

### Official Review · Reviewer_77Fw · 2024-05-25

**Rating:** 3
**Confidence:** 3

**Summary:**

This paper presents a multi-modality framework for action quality assessment. The authors proposed to determine the action quality by integrating skeleton and RGB feature frames, and the skeleton features are enhanced in a self-supervised manner.

**Strengths:**

The paper is well written and organized. Experiments are sufficient and comprehensive to illustrate the effectiveness of the proposed framework.

**Limitations:**

1. The novelty of this paper is limited. Self-supervised learning for skeleton data has been studied and applied in LongTGAN [1] and MS^2L [2].

[1] Unsupervised Representation Learning With Long-Term Dynamics for Skeleton Based Action Recognition. AAAI 2018.

[2] MS^2L: Multi-task Self-supervised Learning for Skeleton Based Action Recognition", ACM MM 2020.

The authors are suggested to elaborate the difference with the previous works in self-supervised learning.
The multi-modality fusion is not new in action analytics [3][4] etc.

[3] Temporal Segment Networks: Towards Good Practices for Deep Action Recognition. ECCV 2016.

[4] Multi-Modality Multi-Task Recurrent Neural Network for Online Action Detection. TCSVT 2020.

2. In Table 1,2,3, the authors are suggested to indicate the modalities each method employs for a fair comparison.

3. In Table 12, it’s interesting to see action recognition accuracy gets improved as mask rate increases when mask rate < 0.5 The authors are suggested to give more explaination on it.

4. Minor: In Eq. (5), y_i is not explained until Eq.(6).

**Suitability:**

2

---

### Meta-Review · Program_Chairs · 2024-07-01

**Recommendation:** Accept (Poster)
**Confidence:** 4

**Metareview:**

We would like to thank the authors for answering the questions by the reviewers in the rebuttal. Reviewers have considered your responses. Reviewers are generally positive towards this paper. Also, some concerns remain: (1) The novelty is limited. (2) Please check the statements and ensure they are precise. (3) Please ensure the comparisons are fair and carefully proofread the paper. This paper is recommended for accept, where authors should address (2) and (3) above in the final version.